# Genetic Diversity and Population Dynamics of *Leptobrachium leishanense* (Anura: Megophryidae) as Determined by Tetranucleotide Microsatellite Markers Developed from Its Genome

**DOI:** 10.3390/ani11123560

**Published:** 2021-12-15

**Authors:** Chao Fu, Qingbo Ai, Ling Cai, Fuyuan Qiu, Lei Yao, Hua Wu

**Affiliations:** Institute of Evolution and Ecology, School of Life Sciences, Central China Normal University, Wuhan 430079, China; fc@mails.ccnu.edu.cn (C.F.); aiqingbo@mails.ccnu.edu.cn (Q.A.); cling@mails.ccnu.edu.cn (L.C.); qiufy@mails.ccnu.edu.cn (F.Q.); yaolei@mails.ccnu.edu.cn (L.Y.)

**Keywords:** Chinese endemic frog, genetic diversity, microsatellite markers, population dynamics, wildlife conservation

## Abstract

**Simple Summary:**

More than 41 percent of amphibians evaluated by International Union for Conservation of Nature are threatened. It is vitally important to establish scientific and effective protection strategies for these organisms. Leishan Spiny Toad is endemic to China and it has a narrow distribution area. Long-term intentional human use and habitat destruction has caused the species to suffer. Here, we developed newly reliable and efficient molecular markers based on its genome to assess its genetic diversity and population history and provided support for conservation of this toad. Our results show that this toad still possesses high genetic diversity, but population decline may increase the possibility of inbreeding, which could work against persisting survival. Recovering the toad’s habitat and strengthening the publicity and education of wildlife protection can be helpful to its sustainability.

**Abstract:**

Persisting declination of amphibians around the world has resulted in the public attaching importance to the conservation of their biodiversity. Genetic data can be greatly helpful in conservation planning and management, especially in species that are small in size and hard to observe. It is essential to perform genetic assessments for the conservation of *Leptobrachium leishanense*, an endangered toad and receiving secondary protection on the list of state-protected wildlife in China. However, current molecular markers with low reliability and efficiency hinder studies. Here, we sampled 120 adult toes from the population in the Leishan Mountain, 23 of which were used to develop tetranucleotide microsatellite markers based on one reference *L. leishanense* genome. After primer optimization, stability detection, and polymorphism detection, we obtained 12 satisfactory microsatellite loci. Then, we used these loci to evaluate the genetic diversity and population dynamics of the 120 individuals. Our results show that there is a low degree of inbreeding in the population, and it has a high genetic diversity. Recently, the population has not experienced population bottlenecks, and the estimated effective population size was 424.3. Accordingly, stabilizing genetic diversity will be key to population sustainability. Recovering its habitat and avoiding intentional human use will be useful for conservation of this species.

## 1. Introduction

Amphibians have long been declining on a global scale, and this trend will continue [1]. Furthermore, some amphibians face extinction or have become extinct [2]. There have been reports of massive declines in amphibians in many places, including areas where all species have been actively conserved [3,4]. Although there has been little consensus on the causes of this phenomenon [5], we recognize that amphibian populations are under serious threat and are in desperate need of conservation.

The Leishan Spiny Toad (*Leptobrachium leishanense*) is an endemic amphibian to China and is mainly restricted in Leishan county of Guizhou Province. This species inhabits broadleaf forests at elevations ranging from 1100–1800 m and breeds in slow-flowing streams via larval development [6]. The toad suffers from significant habitat loss and is often harvested for local consumption [7]. Thus, the population size has declined dramatically. It is listed as an endangered species on the International Union for Conservation of Nature (IUCN) Red List and receives secondary protection on the new list of state-protected wildlife in China. Formulating scientific conservation strategies is necessary for this species.

Genetic assessment is one of the aims of the conservation of biodiversity [8] and an important measure for amphibian population conservation [9,10,11]. Estimating genetic diversity and effective population size are the main goals of genetic assessments [12]. Genetic diversity reflects the adaptive potential of populations for environmental change [13]. When genetic diversity decreases, the extinction risk of populations increases [14]. Moreover, the levels of genetic diversity are related to population size [15]. It is a consensus that determining effective population size is more vital than measuring census size in populations [16]. In theory, small populations are susceptible to genetic depletion through drift and inbreeding, with adverse consequences for viability [17,18]. Therefore, effective population size can be used to assess the viability of populations.

As next-generation sequencing technologies offer new opportunities for conservation genetics [19], microsatellite markers with high mutation rates and genome-wide distributions reveal recent changes in genetic structure and demography critical for population management [20,21]. Although several studies using 10 dinucleotide microsatellites have shown that *L. leishanense* has high levels of genetic diversity and has not experienced recent bottleneck events [22,23,24], genetic assessments of this species are not nearly sufficient. In addition, dinucleotide microsatellites are considered less efficient and more unreliable than tetranucleotides because of their minimal PCR stutter [25]. Moreover, the traditional methods of microsatellite isolation and characterization are quite involved, costly, and time-consuming [26]. With the publication of a number of genomes, we can obtain sufficient numbers of different types of useful microsatellite loci more efficiently [27]. The genome sequencing project of *L. leishanense* has provided the opportunity to isolate and characterize microsatellites at the genomic level [28].

Here, we totally sampled 120 adult toes of *L. leishanense* from the population in the Leishan Moutain, and 23 of them were used to develop tetranucleotide microsatellite markers with polymorphisms based on one reference *L. leishanense* genome. After that, we analyzed the genetic diversity and population dynamics using the microsatellite loci we identified. The goals of this study were to (1) develop microsatellite loci with high reliability and efficiency, (2) evaluate the genetic diversity of the *L. leishanense* population, (3) detect if the population is experiencing a population bottleneck, (4) estimate the effective population size, and (5) provide molecular support for *L. leishanense* conservation planning.

## 2. Materials and Methods

### 2.1. Sampling

In 2012, 2013, 2014, 2015, and 2018, we collected 24 *L. leishanense* adults per year in Maoping village of Leishan County, Guizhou, China (Figure 1), sampled their toes, fixed the toes in anhydrous ethanol, and stored them in a −20 °C refrigerator. All individuals were released immediately after sampling. All experiments involving animals were approved by the Animal Ethics Committee of the School of Life Sciences, Central China Normal University (CCNU-IACUC-2019–008). We have complied with all relevant ethical regulations for animal testing and research.

### 2.2. DNA Extraction and Primer Selection

DNA samples were extracted using the TIANamp DNA kit (Tiangen, Beijing, China) and stored at −20 °C. MicroSatellite identification tool (MISA-web, Gatersleben, Germany) [29] was used to obtain the simple sequence repeats (SSRs) of *L. leishanense* from its genome. Then, we randomly selected 87 tetrabase repeat microsatellite markers that were repeated more than 10 times and designed 87 pairs of primers according to the flanking sequences at both ends of each primer. With the extracted DNA as a template, we optimized the annealing temperature of the primers and reaction system. Each polymerase chain reaction (PCR) procedure was conducted in a 10 μL volume, in which the premix was 5 μL, each primer was 0.3 μL, template DNA was 0.6 μL, and ddH_2_O was 3.8 μL. The procedure was performed with initial denaturation at 94 °C for 5 min, followed by 35 cycles of denaturation at 94 °C for 30 s, annealing at temperature Ta for 30 s, extension at 72 °C for 45 s, and extension at 72 °C for 5 min. PCR products were detected by 1% agarose gel electrophoresis. By adjusting the Ta temperature, a product with a clear band was obtained. The Ta temperature corresponding to the product was used as the optimum temperature for PCR amplification. Under the optimal amplification conditions, we used the DNA of three different individuals to detect the stability of primers in different individuals and screened the primers that could be amplified stably.

### 2.3. Polymorphic Microsatellite Verification

The screened primers were used to synthesize 5′ upstream fluorescent primers (FAM, HEX and TEMED, compounded by Tiangen, Beijing, China). DNA amplification was performed on 23 individuals collected in 2012 and 2013 by PCR with fluorescent primers, and the amplified fluorescence PCR products were sent to Tsingke Biological Company, Beijing, China for SSR scanning and sequenced by an ABI 3730xl analyzer. Then, the products were genotyped and calculated, and the evaluation criterion of the polymorphisms was a *PIC* value higher than 0.5 [30]. We used Genemarker 1.3 software [31] to read the lengths of alleles, genotyped the microsatellite markers, and selected the sites with obvious polymorphisms for the following analysis. The microsatellite genotyping data in Excel were transformed by using the Microsatellite Toolkit [32]. Cervus 3.0 software [33] was used to calculate the number of alleles (Na), polymorphism information content (*PIC*), expected heterozygosity (*He*), and observed heterozygosity (*Ho*). Micro-Checker 2.2.3 [34] was used to check large allele dropout of the microsatellite markers. GenePop 1.2 software [35] was used to detect the Hardy–Weinberg equilibrium (HWE) and linkage disequilibrium (LD) of the screening microsatellite markers with polymorphisms, and the Bonferroni correction was used for correction. The significance level was *p* < 0.05.

### 2.4. Genetic Diversity Analysis

A total of 120 DNA samples were amplified by PCR with the screening fluorescent primers described above. The PCR products were sent to Qingke Biological Company for SSR scanning. An ABI 3730xl analyzer was used for sequencing. Data were analyzed using GenAlEx 6.502 [36] to calculate the effective number of alleles (Ne), the mean relatedness of the individuals for every year, and the per year genetic differentiation coefficient (*Fst*), and Cervus 3.0 software was used again to calculate the values described above. Excel and Microsatellite Toolkit v3.1.1 software were used for preliminary genetic data statistics and data format conversion. FSTAT 2.9.3.2 software [37] was used to calculate allelic richness (Ar), allelic diversity (Hs), and the inbreeding coefficient (*Fis*).

### 2.5. Population Bottleneck Identification

Bottleneck 1.2.02 software [38] was used to test whether the population had experienced population bottlenecks. Sign and Wilcoxon methods were used to test mutations through three mutation models: the infinite allele model (IAM), the stepwise mutation model (SMM), and the two-phased model of mutation (TPM). TPM was set to 95% SMM, with a variance of 30 and 1,000 iterations.

### 2.6. Effective Population Size Calculation

NeEstimator 2.1 [39] was used to calculate the effective population size by selecting the random mating model, and the confidence interval was 95%.

## 3. Results

### 3.1. Distribution of SSR in Genome of L. leishanense

A total of 1,454,145 microsatellite markers were obtained from the genome of *L. leishanense*. Monobase repeat microsatellite markers and dibase repeat microsatellite markers were the most common among all microsatellite markers. There were 874,773 monobase repeat microsatellite markers and 263,927 dibase repeat microsatellite markers, accounting for 60.16% and 18.15% of the total number of microsatellite markers, respectively, followed by 71,167 tribase repeat microsatellite markers and 23,332 tetrabase repeat microsatellite markers, accounting for 4.89% and 1.60% of the total number of microsatellite markers, respectively. The number of pentabase repeat microsatellite markers and hexabase repeat microsatellite markers was the lowest, with 909 pentabase repeat microsatellite markers and 844 hexabase repeat microsatellite markers, accounting for only 0.12% of the total microsatellite markers (Figure 2).

### 3.2. Polymorphism Microsatellite Loci

Eighty-seven pairs of primers randomly chosen from 23,332 tetrabase repeat microsatellite marker. After primer optimization, 64 pairs of primers were successfully amplified. Then stability detection was used, and 46 pairs of primers were obtained. Employing polymorphism detection, we obtained 12 satisfactory microsatellite loci. The Na of these loci ranged from 6–16. *PIC* values ranged from 0.537–0.904. *Ho* and *He* were between 0.609–0.913 and between 0.622–0.931, respectively (Table 1). All 12 loci were not significant with regard to LD (*p* > 0.05), and there were no loci deviated from HWE (*p* > 0.05). According to Micro-Checker 2.2.3, there was no large allele dropout of these microsatellite markers and no scoring error caused by the shadow peak.

### 3.3. Population Genetic Diversity

Using the 12 loci screened above, all 120 individuals were used to study the genetic diversity of the population. HWE detection of the *L. leishanense* population was performed. After Bonferroni correction, the significance level was *p* < 0.0042. The results are shown in Table 2. When we used all 120 samples for testing, loci *LEA23*, *LEA7, LEA47*, *LEA2,* and *LEA53* deviated from HWE significantly, with *Fis* values as 0.179, 0.112, 0.119, 0.230, and 0.134, respectively (Table 3). When we separated the samples into each year for testing, locus *LEA20* deviated from HWE significantly in 2013, with a *Fis* value of 0.215. Locus *LEA23* and *LEA7* deviated from HWE significantly in 2015 and 2018. *Fis* values of locus *LEA23* in 2015 and 2018 are 0.132 and 0.249, respectively. *Fis* values of locus *LEA7* in 2015 and 2018 are 0.161 and 0.369, respectively. Locus *LEA47* deviated from HWE significantly in 2018 with *Fis* value as 0.201, and locus *LEA2* deviated from HWE significantly in 2015 with *Fis* value as 0.218.

Then, we calculated pairwise year *Fst* valus in *L. leishanense* (Table 3). None of these values is greater than 0.05, suggesting the genetic differentiation between these years is negligible [40]. Further, we calculated the mean relatedness of the individuals for every year (Figure 3). In 2013, 2014, and 2015, mean pairwise relatedness within groups was significantly greater than zero, indicating the samples we collected in these three years have relatively close relationships.

Next, we calculated Na, Ne, *PIC*, *Ho*, *He*, Ar, Hs, and *Fis* of the population (Table 4). The results indicated that the genetic diversity of the population was still high. The positive value of *Fis* and that of *Ho* was lower than that of *He*, suggesting that there was a low degree of inbreeding in the population.

### 3.4. Population Bottleneck

The average expected heterozygosity (*Heq*) of the population in the IAM, SMM, and TPM models was calculated (Table 5). In the IAM model, there were 11 sites where He was significantly higher than Heq (*p* < 0.05), among which *LEA22*, *LEA25*, *LEA20*, *LEA35*, *LEA14*, *LEA23,* and *LEA24* were extremely significantly higher than Heq (*p* < 0.01). In the TPM and SMM models, only *He* at *LEA5* was significantly higher than *Heq*, showing heterozygote surplus. The sign test and Wilcoxon test were used to detect the heterozygosity surplus of the population under the three models of IAM, TPM, and SMM (Table 6). The mutation-drift balance of the population was detected under the IAM model, both of the sign and Wilcoxon tests showed significant deviations from the mutation-drift balance of the population. Under the TPM and SMM models, both the results of the sign test and Wilcoxon test showed that the population did not deviate from mutation-drift equilibrium.

The analysis of allele frequency distribution in the *L. leishanense* population showed that the allele frequency was mainly concentrated between 0.0–0.1, which was approximately 84.52% of the total allele frequency. Alleles with a frequency of 0.1–0.2 accounted for 12.69% of the total allele frequency. The proportion of the frequency distribution interval of 0.2–0.3 was 1.80%, while the allele proportions of the frequency distribution intervals of 0.3–0.4 and 0.4–0.5 were 0.5%. The allele frequency showed a typical “L” type distribution (Figure 4), suggesting that the population has not recently experienced a bottleneck effect.

### 3.5. Effective Population Size

According to the LD distance, the effective population size was estimated to be 424.3 (95% CI = 272.7–878.2) from NeEstimator 2.1.3.1.

## 4. Discussion

We isolated and characterized 12 tetrabase repeat microsatellite markers with polymorphisms from one reference genome of *L. leishanense*. Then, we used these loci to study the genetic diversity and population dynamics of this species. We found that the genetic diversity of the population was high and that there was a low degree of inbreeding in the population. Moreover, the population has not recently experienced bottleneck effects, and the estimated effective population size is 424.3.

### 4.1. Tetranucleotide Microsatellite Markers

Although the results above are similar to those of Zhang’s research [24], which used 10 dibase repeat microsatellite markers, the 12 tetranucleotide microsatellite markers we developed are more polymorphic and suitable for genetic diversity research. During PCR amplification, a biological phenomenon called stutter is generated due to chain slippage, resulting in typing errors, and the stutter product has one or more fewer duplicates than the real allele product [41,42]. In general, tetranucleotide repeats tend to stutter less than trinucleotide and dinucleotide repeats and are much more accurate and reliable [43,44]. Therefore, in different types of microsatellite systems, tetrabase repeat microsatellite markers are more common than dibase or tribase markers. Moreover, the *PIC* values of all 12 loci were higher than 0.5, suggesting that the loci we developed had higher polymorphism. Stable and reliable microsatellite markers are a necessary prerequisite for population estimation in the wild [45]. Thus, after primer optimization, stability detection, and polymorphism detection, we finally obtained 12 satisfactory tetranucleotide microsatellite loci.

### 4.2. Genetic Diversity

We speculated that several loci deviated from HWE (Table 2) mainly caused by sampling from the same family (Figure 3). The sharp decline of the population size has increased the possibility in sampling individuals of same family. The genetic diversity of a population is a long-term process, the population of *L. leishanense* does not have significant genetic differentiation among these five years (Table 3); accordingly, we considered that these deviating loci were still effective in estimating population genetic diversity. Then, a series of indices were used to measure the genetic diversity of the toad, including Na, Ar, Ne, *PIC*, *Ho*, *He*, and Hs. According to our results, the toad still has high genetic diversity. Threatened species usually have small or declining populations and are prone to loss of genetic diversity due to inbreeding or genetic drift [14]. As an endangered and narrowly distributed toad, the population shows the opposite result. Several studies investigating endangered or narrowly distributed species have obtained similar results [45,46,47,48,49], indicating that endangered species or species with a narrow distribution may also have high levels of genetic diversity. When the earth was in an ice age, some areas with a stable ecological environment became the refuge of organisms, and the populations living in the refuge survived and accumulated rich genetic diversity [50]. The Leishan Spiny Toad is a relatively primitive species, and its formation dates back to the Miocene [22]. The toad survived by staying on Leigong Mountain and retained rich genetic diversity when the ice age came. In addition, two additional distribution sites were found by Zheng et al. [51], suggesting that the toad is not strictly a narrowly distributed species. We may have underestimated the genetic diversity of the species.

Although the *Fis* value of the population is on the low degree, this does not mean that there is no inbreeding between the individuals in the population. According to our year-by-year field work, its population size is declining. This undoubtedly increases the possibility of its inbreeding. Inbreeding has a negative effect on the fitness of the population, including fertility and viability [52], which is not conducive to the long-term development of the population. We could not find more obvious molecular evidence of inbreeding, possibly due to our restricted sampling size and the relatively high number of alleles found. As with high number of alleles, the probability of obtaining homozygote hgenotypes in one locus is very low. Thus, it will influence our detection of inbreeding.

### 4.3. Population Dynamics

Combining the results of model simulation with allele frequency distribution, we find that the population has not recently experienced a bottleneck effect. We tested three models, and IAM was significant both in the sign test and the Wilcoxon test. Both SMM and TPM were not significant in the sign test and Wilcoxon test (Table 5). IAM assumes that there is only one mutation of an allele in a population, and each mutation produces a new allele, which is generally used in isozyme or DNA sequencing data. SMM supposes that alleles can mutate upward or downward into new alleles. TPM is the synthesis of the previous two models, and the probability of occurrence of two kinds of mutations can be determined. The principle of allelic mutation in microsatellite data is the increase or absence of repeating units, which is represented by the change in sequence length. Some studies believe that TPM is more suitable for microsatellite data [53]. Therefore, we accept the result of TPM that there is no significant excess heterozygosity in this population. That is, the rate of heterozygosity decrease is approximately the same as the rate of allele loss in *L. leishanense*, indicating that the population has not recently experienced a bottleneck.

However, the ability to detect the population bottleneck based on heterozygosity is limited, and the number of alleles is more sensitive to population fluctuation, so it is more reliable to analyze the distribution of allele rates in the case of heterozygous residues to determine whether the population has experienced the bottleneck effect [54]. To enhance the adaptability to environmental changes, species tend to accumulate many rare alleles with low frequency. Therefore, the frequency distribution of alleles in mutation–drift equilibrium shows an “L” shape. If the species recently experienced a genetic bottleneck, the distribution of alleles with low frequency (0.0–0.1) will change to a mid-frequency distribution (0.1–0.2); thus, the allele frequency distribution will deviate from “L” [55]. In this study, the frequencies of the allele were generally in a typical “L” (Figure 3), suggesting that the population did not experience bottleneck effects.

The LD distance between microsatellite markers can be used to estimate effective population size, and this method has been applied to mammals, fish, amphibians, and other animals [56]. Effective population size is a valuable method in population conservation and management research. Maintaining an effective population of sufficient size is a key factor to maintain the rich genetic diversity of the population. Based on the microsatellite loci we developed, the estimated effective population size of *L. leishanense* is 424.3. Nei et al. [57] deemed that the population size should be 4–10 times of the effective population size to maintain the stability of population genetic diversity. Therefore, to maintain the stability of the population, the number of Leishan Spiny Toads should be 1697.2–4243. However, while LD information is used to estimate the effective population size, the accuracy of the results is significantly correlated with the sample size [58]. More samples may be needed to obtain more reliable and accurate results in *L. leishanense*.

## 5. Conclusions

Our study has provided 12 reliable tetranucleotide microsatellite loci with polymorphisms, enriching the information regarding the genetic diversity and population dynamics of *L. leishanense*. Although the genetic diversity is still high based on our results, a low degree of inbreeding indicates that the population is declining. Avoiding habitat fragmentation and intentional human use will be key to the conservation of this species. Furthermore, recovering the streams and woodlands where the species once existed abundantly will also help to stabilize its genetic diversity.

## Figures and Tables

**Figure 1 animals-11-03560-f001:**
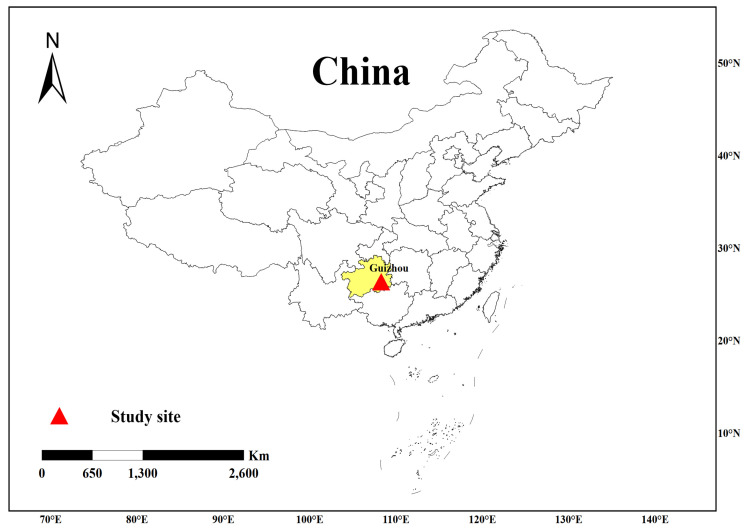
Sampling site of *L. leishanense*. The red triangle represents the sampling site: Maoping village of Leishan County, Guizhou Province, China.

**Figure 2 animals-11-03560-f002:**
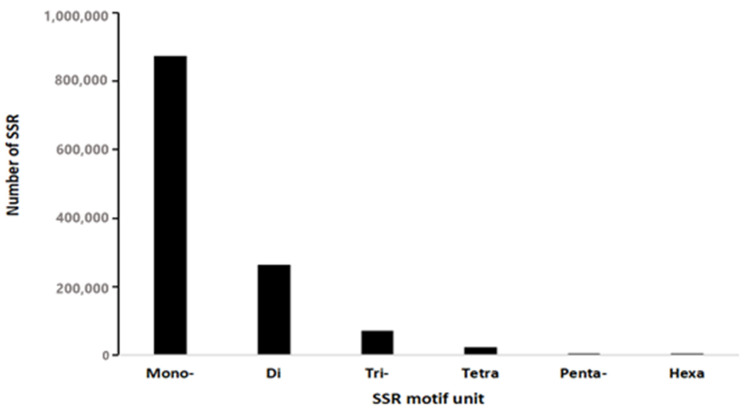
Distribution of SSR repeat types in the genome of *L. leishanense*.

**Figure 3 animals-11-03560-f003:**
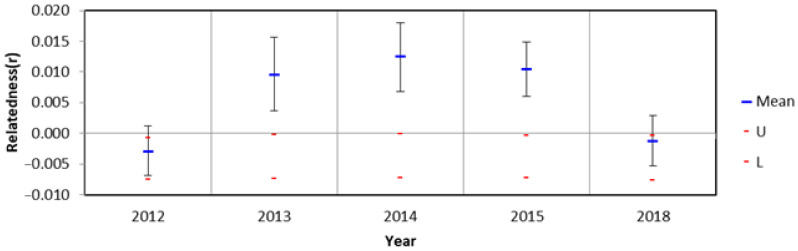
Mean within-year pairwise relatedness estimates for 5 years in *L. leishanense*. Red lines represent permuted 95% confidence intervals around the null hypothesis of zero relatedness and error bars represent bootstrapped confidence intervals around the mean.

**Figure 4 animals-11-03560-f004:**
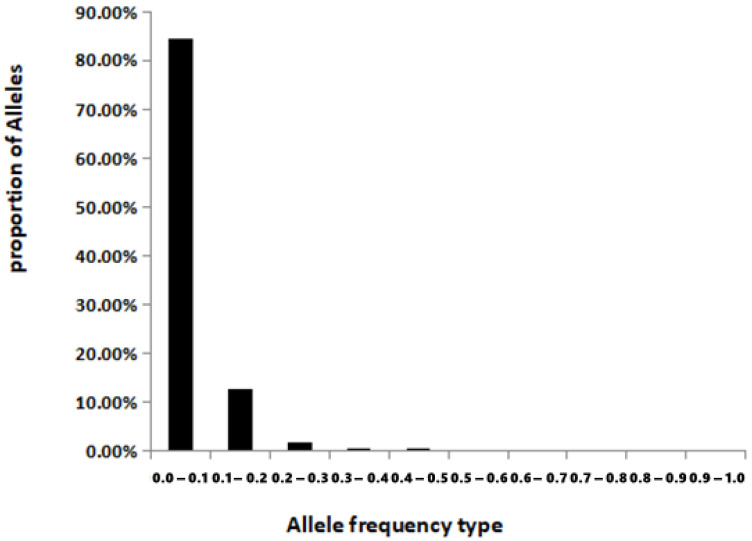
Allele frequency distribution in the population of *L. leishanense*.

**Table 1 animals-11-03560-t001:** Detailed information on 12 microsatellite loci developed from 23 *L. leishanense*.

Locus	Primer Sequence (5′-3′) (F, Forward; R, Reverse)	Repeat Motif	PCR Production (μL)	Labelling Dye	Ta (°C)	Size Range (bp)	Na	*Ho*	*He*	*PIC*
LEA22	F:TGCGACTACGTAACCCTGTG R:AGGAAATGAGCCTTTGCCTC	(AGAT)_16_	3	5′ FAM	56 °C	216–292	6	0.609	0.622	0.537
LEA25	F:GTGGTTGGTTGGTTGGGTC R:TGGTCAGGATGTGAGGAGTG	(TGGT)_13_	6	5′ HEX	58 °C	226–282	14	0.826	0.889	0.856
LEA20	F:ATTTGATGGTGTCTGGGAGG R:CTAAGAGAGCCGAAACGTCG	(GATA)_13_	9	5′ TAMRA	58 °C	195–263	16	0.870	0.930	0.903
LEA35	F:GCGGGAGTTTGAGCTGTATC R:CAGCTTACATTGTGTGCAGC	(CTAT)_14_	3	5′ FAM	62 °C	192–260	16	0.913	0.925	0.897
LEA14	F:ATAAGCTAAACAGGCGTGGG R:TTTCATATCAGGGGAGAGCG	(TTTC)_18_	6	5′ HEX	62 °C	150–234	14	0.870	0.882	0.851
LEA23	F:CCAGGAACAAGGTCAGTGGT R:CCCATGTTCGAGAGGAGAAG	(TCTA)_18_	9	5′ TAMRA	64 °C	178–258	10	0.739	0.850	0.812
LEA5	F:TCAACTCAACTCTCCCCCTG R:AACGCACATCCCTAGTGGTC	(CTTT)_14_	3	5′ FAM	60 °C	183–199	11	0.826	0.859	0.823
LEA7	F:ACCATCAATTTTAGGGGTGC R:TGGGATTTCCCAGTCATTTC	(AGAT)_20_	6	5′ HEX	60 °C	178–246	14	0.826	0.908	0.879
LEA47	F:GACAAATGGGGAGATGATGG R:AAAACGTCAGTGGCAAATCC	(AGAT)_17_	9	5′ TAMRA	62 °C	162–261	11	0.739	0.886	0.853
LEA24	F:GTGAAACTTGCATCCACTGC R:AAAATTAGCTATGGGTGGCG	(TATC)_20_	3	5′ FAM	62 °C	205–289	15	0.913	0.931	0.904
LEA2	F:CACCCCGTGACAATATACCC R:TGAGGGATCATTCTTCTGGC	(GATA)_11_	6	5′ HEX	62 °C	207–251	11	0.913	0.892	0.859
LEA53	F:ATGGATAGATGGATGGCTGG R:CAACGCGGAAAAAGAAACAT	(TAGA)_13_	9	5′ TAMRA	62 °C	210–254	13	0.913	0.918	0.889

**Table 2 animals-11-03560-t002:** Hardy–Weinberg equilibrium test of 12 loci in *L. leishanense*.

Locus	2012–2018	2012	2013	2014	2015	2018
	*P* _HWE_	*P* _HWE_	*P* _HWE_	*P* _HWE_	*P* _HWE_	*P* _HWE_
LEA22	0.061	0.441	0.073	0.157	0.282	0.031
LEA25	0.044	0.737	0.009	0.139	0.509	0.073
LEA20	0.019	1.000	0.003*	0.541	0.205	0.335
LEA35	0.435	0.229	0.607	0.207	0.917	0.151
LEA14	0.070	0.266	0.025	0.479	0.395	0.091
LEA23	0.000 *	0.108	0.023	0.038	0.000 *	0.003 *
LEA5	0.237	0.657	0.053	0.843	0.984	0.208
LEA7	0.004 *	0.541	0.330	0.426	0.004 *	0.004 *
LEA47	0.000 *	0.862	0.100	0.695	0.014	0.004 *
LEA24	0.037	0.133	0.914	0.844	0.447	0.086
LEA2	0.000 *	0.006	0.010	0.012	0.000 *	0.008
LEA53	0.001 *	0.017	0.280	0.491	0.727	0.019

Note: * Indicated significant deviations from Hardy–Weinberg equilibrium after Bonferroni correction.

**Table 3 animals-11-03560-t003:** Pairwise year *Fst* values in *L. leishanense*.

Year	2012	2013	2014	2015	2018
2012	0.000	0.006	0.010	0.011	0.000
2013	0.006	0.000	0.010	0.032	0.017
2014	0.010	0.010	0.000	0.038	0.019
2015	0.011	0.032	0.038	0.000	0.009
2018	0.000	0.017	0.019	0.009	0.000

**Table 4 animals-11-03560-t004:** Genetic diversity indices of *L. leishanense* in 120 individuals.

Locus	Na	Ar	Ne	*PIC*	*Ho*	*He*	Hs	*Fis*
LEA22	25.000	25.000	14.371	0.926	0.858	0.934	0.935	0.082
LEA25	16.000	16.000	9.658	0.888	0.792	0.900	0.901	0.121
LEA20	21.000	21.000	13.097	0.919	0.867	0.928	0.928	0.066
LEA35	19.000	19.000	11.950	0.910	0.908	0.920	0.920	0.013
LEA14	16.000	16.000	10.119	0.893	0.858	0.905	0.905	0.052
LEA23	23.000	23.000	14.180	0.925	0.767	0.933	0.934	0.179
LEA5	11.000	11.000	3.034	0.615	0.583	0.673	0.674	0.134
LEA7	17.000	17.000	8.177	0.867	0.783	0.881	0.882	0.112
LEA47	25.000	25.000	12.991	0.918	0.817	0.927	0.927	0.119
LEA24	20.000	20.000	12.010	0.911	0.867	0.921	0.921	0.059
LEA2	16.000	16.000	8.518	0.873	0.683	0.886	0.887	0.230
LEA53	12.000	12.000	7.234	0.847	0.750	0.865	0.866	0.134
Mean	18.417	18.417	10.445	0.874	0.794	0.890	0.890	0.107

Notes: number of alleles (Na), allelic richness (Ar), effective number of alleles (Ne), polymorphism information content (*PIC*), observed heterozygosity (*Ho*), expected heterozygosity (*He*), allelic diversity (Hs), and the inbreeding coefficient (*Fis*).

**Table 5 animals-11-03560-t005:** Bottleneck test of 12 microsatellite loci in *L. leishanense*.

Locus	Sample Size	*He*	IAM		TPM		SMM	
			*Heq*	*p*	*Heq*	*p*	*Heq*	*p*
LEA22	120	0.934	0.877	0.005 *	0.930	0.453	0.937	0.336
LEA25	120	0.900	0.792	0.004 *	0.885	0.296	0.899	0.473
LEA20	120	0.928	0.850	0.000 *	0.914	0.228	0.923	0.502
LEA35	120	0.920	0.829	0.002 *	0.902	0.176	0.916	0.501
LEA14	120	0.905	0.796	0.003 *	0.884	0.186	0.899	0.456
LEA23	120	0.933	0.864	0.001 *	0.923	0.290	0.933	0.495
LEA5	120	0.673	0.700	0.318	0.825	0.010 *	0.848	0.000 *
LEA7	120	0.881	0.808	0.080	0.893	0.260	0.905	0.091
LEA47	120	0.927	0.876	0.028 *	0.930	0.371	0.937	0.144
LEA24	120	0.921	0.836	0.007 *	0.910	0.332	0.920	0.448
LEA2	120	0.886	0.793	0.030 *	0.884	0.476	0.900	0.205
LEA53	120	0.865	0.727	0.018 *	0.844	0.296	0.862	0.469

Notes: * Significant difference between *He* and *Heq* (*p* < 0.05).

**Table 6 animals-11-03560-t006:** Bottleneck significance test on populations of *L. leishanense*.

Test	*P* _IAM_	*P* _TPM_	*P* _SMM_
Sign test	0.0230 *	0.1887	0.5983
Wilcoxon test	0.0002 *	0.0549	0.7651

Notes: * Significant deviation from mutation-drift equilibrium at *p* < 0.05.

## Data Availability

The datasets generated and analyzed during the current study are available from the corresponding author on reasonable request.

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
