# Peer review of "Genetic Diversity and Population Dynamics of Leptobrachium leishanense (Anura: Megophryidae) as Determined by Tetranucleotide Microsatellite Markers Developed from Its Genome"

_animals, 2021, doi:10.3390/ani11123560_

Round 1

Reviewer 1 Report

Although the authors used appropriate analytical and statistical methods for achieving this goal, the manuscript needs clarification in few points. 

Sampling seems to be an issue. For such a study, the number of the individuals examined, 120 individuals spread in 5 years, combined with the use of only one genetic marker, probably, preventes, in several occasions, the authors to conclude safely. Therefore, many of the conclusions lack a strong statistical support. I would like from the authors a more substantiated argument for the lack of more individuals. 

In addition, I did not see any reference to or comparison with the other study conducted by Wei ZHANG,Zhenhua LUO,Mian ZHAO,Hua WU that also deals with the same subject.

In Conlusions section p. 5 line 324:

"Although the genetic diversity is still high based on our results, a low 
degree of inbreeding indicates that the population is declining."

Why? I would like a more convincing argument for this statement.

Author Response

Dear reviewer,

Thanks for your kindly comments and suggestions. We have tried our best to rewrite and improve our manuscript and streamlined the manuscript in this version. Below, we list your comments verbatim along with our response in bold, detailing how we addressed the suggestions and criticisms.

Point 1: For such a study, the number of the individuals examined, 120 individuals spread in 5 years, combined with the use of only one genetic marker, probably, preventes, in several occasions, the authors to conclude safely. Therefore, many of the conclusions lack a strong statistical support. I would like from the authors a more substantiated argument for the lack of more individuals.

Response 1: We do realize the sampling size could make a difference in our results. We increased calculating the mean relatedness of the sampling individuals for every year and pairwise year Fst values, then reviewed our manuscript in ‘3.3’ and ‘4.2’ (Line 217-227, 324-332). Our field work on L. leishanense shows that its population size is declining, which makes it difficult for us to collect more samples. Certain studies on the effect of sample size on population genetic diversity analysis have suggested that a minimum sample size of 20-30 may be appropriate when using microsatellite markers for population genetic structure analysis (Yan et al. 2004. Effects of sample size on various genetic diversity measures in population genetic study with microsateilite DNA markers. Acta Zool Sin, 50(2), 279-290; Liu et al. 2005. Study on the Optimal Sample and Loci Quantity for RAPD Analysis. J Zhanjiang Ocean Univ, 25(4), 1-4; Lu et al. 2008. Effects of sample size on various genetic structure parameter in cultured population genetic study. J Fish Chn, 32(5), 674-683). Therefore, we believe that most of our results are stable despite our small sample size. But there's still a problem, although its population size is declining, we could not find significant molecular evidence of inbreeding. This may be due to our limited sample size, but relatively more alleles were found. As with high number of alleles the probability of getting homozygote hgenotypes in one locus is very low. Thus, it will influence our detection of inbreeding.

Point 2: In addition, I did not see any reference to or comparison with the other study conducted by Wei ZHANG, Zhenhua LUO, Mian ZHAO, Hua WU that also deals with the same subject.

Response 2: We referred Zhang’s research in our ‘Introduction’ in Line 67-69, which used 10 dinucleotide microsatellites and showed that L. leishanense has high levels of genetic diversity and has not experienced recent bottleneck events. This results are similar to our research. But the 12 tetranucleotide microsatellite markers we developed are more polymorphic and suitable for genetic diversity research (Line 285-287).

Point 3: In Conlusions section p. 5 line 324:"Although the genetic diversity is still high based on our results, a low degree of inbreeding indicates that the population is declining."Why? I would like a more convincing argument for this statement.

Response 3: We increased calculating the mean relatedness of the sampling individuals for every year and pairwise year Fst values. We added the explanation for low degree of inbreeding and population decline in ‘4.2’ like ‘According to our year-by-year field work, its population size is declining. This undoubtedly increases the possibility of its inbreeding. Inbreeding has a negative effect on the fitness of the population, including fertility and viability, which is not conducive to the long-term development of the population. We could not find more obvious molecular evidence of inbreeding may be due to our restricted sampling size and the relatively high number of alleles found. As with high number of alleles the probability of getting homozygote hgenotypes in one locus is very low. Thus, it will influence our detection of inbreeding.’ (Line 324-332).

Best regards,

Prof. Wu Hua

Reviewer 2 Report

This paper present the genetic diversity and population dynamics of Leishan Spiny Toad (Leptobrachium leishanense) with 12 tetranucleotide microsatellite markers, which were developed based on its reference genome. It is important to evaluate the population genetic diversity and dynamics of such endangered species, for it is valuable to to determine the conservation status of endangered species. I think the data is adequate and the analyses are robust to the question. On the whole, I think this manuscript is well written, except that English needs further improvement. My suggestion is that it could be accepted after some modification, here I listed some comments as following for improving the manuscript.

Line 16. Change the verb ‘showed’ to ‘show’, and change ‘gets’ to ‘possesses’.

Line 26. ‘120 adult toes’ are they from the same population? Please clarify.

Line 46. Since you mentioned that this species also occur in other areas, here the word ‘only’ is inaccurate, I suggest changing to ‘mainly restricted in the Leishan county...’.

Line 49. ‘polulation’ is a typo, revise it.

Line 50. Add ‘species’ after ‘Endangered’.

Line 65. What does ‘short-term changes’ mean? Do you want to say ‘recent changes’? please revise it.

Line 76. If all of these 120 samples were sampled from one population, you may need to rephrase this sentence: we totally sampled 120 adult toes of L. leishanense from the population in the Leishan Mountain.

Line 89. Add an ethical statement after the description of sampling.

Line 91. Make sure the map of China is accurate.

Line 99. Revised to ‘annealing temperature of the primers and reaction system’.

Line 110. Change ‘Polymorphism’ to ‘Polymorphic’.

Line 114. SSR or STR? Using the same shortage, and provide the full name when first use it.

Line 115. Add the reference citation for each software.  

Line 141. Have you calculated the LD distance of L. leishanense? Please explain how to calculate Ne based on LD distance.

Line 143. The Results section has some redundant description on methods, you may need to state the results, with minimal explanation on methods.

Line 145. How to get these microsatellite markers from the genome data? This need to be clarified in the method section.

Line 160. ‘Eighty-seven’.

Line 169. Please specify clearly of the reverse primer sequence of the locus LEA53 in Table 1, which used ‘-’ as the last base. In general, A or T base wasn't used as the 3’ end of a primer, the 3’ end of the reverse primer sequence of the locus LEA53 has 5 A base.

Line 175. ‘Genetic’ to ‘genetic ’.

Line 180. Delete ‘very’.

Line 198. You may use the “population” as the object instead of the 120 individuals.

Lines 201-202. What is Na? Why Na>Ne is caused by inbreeding? What does the unevenly distributed alleles in populations mean? Please explain it.

Line 203. Provide a statement on the full names of all these statistics at the end of the table.

Line 213. State the significant level is enough. You may write as: both of the sign and Wilcoxon tests showed significant deviations from the mutation-drift balance of the population. By the way, what does this deviation mean? A sign of bottleneck? Please explain it.

Line 244. Change ‘who’ to ‘which’.

Lines 260-261. This sentence makes no sense, I suggest deleting it.

Lines 278-280. I was confused, so you used two methods to evaluate inbreeding? What is the exact inbreeding level of this species?

Line 310. Delete ‘extremely’. 

In addition, I suggest that the author can try to use MsVar to analyze population dynamics, and there may be more substantial results. A

Author Response

Dear reviewer,

Thanks for your kindly comments and suggestions. We have tried our best to rewrite and improve our manuscript and streamlined the manuscript in this version. Below, we list your comments verbatim along with our response in bold, detailing how we addressed the suggestions and criticisms.

Point 1: Line 16. Change the verb ‘showed’ to ‘show’, and change ‘gets’ to ‘possesses’.

Response 1: We have revised them in the manuscript. (Line 16)

Point 2: Line 26. ‘120 adult toes’ are they from the same population? Please clarify.

Response 2: We collected all samples from the population in the Leishan Mountain. Now we change our sentence to ‘Here, we sampled 120 adult toes from the population in the Leishan Mountain’. (Line 26)

Point 3: Line 46. Since you mentioned that this species also occur in other areas, here the word ‘only’ is inaccurate, I suggest changing to ’mainly restricted in the Leishan county...’.

Response 3: We change our sentence to ‘The Leishan Spiny Toad (Leptobrachium leishanense) is an endemic amphibian to China and is mainly restricted in Leishan county of Guizhou Province’. (Line 46)

Point 4: Line 49. ’polulation’ is a typo, revise it.

Response 4: We have revised it in the manuscript. (Line 49)

Point 5: Line 50. Add ’species’ after ’Endangered’.

Response 5: We have revised it in the manuscript. (Line 50)

Point 6: Line 65. What does ’short-term changes’ mean? Do you want to say ’recent changes’? please revise it.

Response 6: We have revised it in the manuscript. (Line 66)

Point 7: Line 76. If all of these 120 samples were sampled from one population, you may need to rephrase this sentence: we totally sampled 120 adult toes of L. leishanense from the population in the Leishan Mountain.

Response 7: We change our sentence to ‘Here, we totally sampled 120 adult toes of L. leishanense from the population in the Leishan Moutain’. (Line 77-78)

Point 8: Line 89. Add an ethical statement after the description of sampling.

Response 8: We add an ethical statement in the manuscript like ‘All experiments involving animals were approved by the Animal Ethics Committee of the School of Life Sciences, Central China Normal University (CCNU-IACUC-2019–008). We have complied with all relevant ethical regulations for animal testing and research.’. (Line 91-94)

Point 9: Line 91. Make sure the map of China is accurate.

Response 9: We reworked our map in the manuscript. (Line 96)

Point 10: Line 99. Revised to ’annealing temperature of the primers and reaction system’.

Response 10: We have revised them in the manuscript. (Line 106-107)

Point 11: Line 110. Change ’Polymorphism’  to ’Polymorphic’.

Response 11: We have revised it in the manuscript. (Line 118)

Point 12: Line 114. SSR or STR? Using the same shortage, and provide the full name when first use it.

Response 12: We change ‘STR scanning’ to ‘SSR scanning’. The first time ‘SSR’ appearance is in Line 101. (Line 122)

Point 13: Line 115. Add the reference citation for each software.

Response 13: We add the reference citation for all software. (Line 125-152)

Point 14: Line 141. Have you calculated the LD distance of L. leishanense? Please explain how to calculate Ne based on LD distance.

Response 14: We do not calculated the LD distance of L. leishanense separately. We mentioned it in the manuscript just because it is the basic principle of NeEstimator 2.1 software. Perhaps we state it not clear and inaccurate. This software calculate Ne based on genotype linkage disequilibrium and heterozygote surplus, we obtained the input file from FSTAT and Genepop software. After choosing the model and the confidence interval, NeEstimator would output the Ne. Now we delete the redundant description in our manuscript. (Line 152)

Point 15: Line 143. The Results section has some redundant description on methods, you may need to state the results, with minimal explanation on methods.

Response 15: We change the statement of ‘3.1’ and ‘3.2’ , delete some redundant description, and remove some explanation on methods. (Line 123-124, Line 167-169, Line 172-193)

Point 16: Line 145. How to get these microsatellite markers from the genome data? This need to be clarified in the method section.

Response 16: We add the method in ‘2.2’. ‘MicroSatellite identification tool (MISA) (29) was used to obtain the simple sequence repeats (SSRs) of L. leishanense from its genome. ‘ (Line 101-103)

Point 17: Line 160. ’Eighty-seven’.

Response 17: We have revised it in the manuscript. (Line 185)

Point 18: Line 169. Please specify clearly of the reverse primer sequence of the locus LEA53 in Table 1, which used ‘-’ as the last base. In general, A or T base wasn't used as the 3’ end of a primer, the 3’ end of the reverse primer sequence of the locus LEA53 has 5 A base.

Response 18: This is a typographical error. We have reduced the font size of the primer sequence. (Line 194)

Point 19: Line 175. ‘Genetic’ to ‘genetic ’.

Response 19: We have revised it in the manuscript. (Line 196)

Point 20: Line 180. Delete ‘very’.

Response 20: We have revised it in the manuscript. (Line 201)

Point 21: Line 198. You may use the ‘population’ as the object instead of the 120 individuals.

Response 21: We have revised it in the manuscript. (Line 234-235)

Point 22: Lines 201-202. What is Na? Why Na>Ne is caused by inbreeding? What does the unevenly distributed alleles in populations mean? Please explain it.

Response 22: Na is the number of alleles. Ne is the effective number of alleles. Na is checked by Cervus software from our microsatellite data and Ne is a theoretical value calculated from homozygosity. The closer the values are, the more evenly the alleles are distributed in the population (Hines et al. 1981. Linkage relationships among loci of polymorphisms in blood and milk of cattle. Journal of Dairy Science, 64(1), 71-76.). The difference between these two values may be caused by the unevenly distribution of alleles, or it may be caused by the restriction of sampling size (Li et al. 2006. Analysis of genetic variation of abalone (haliotis discus hannai) populations with microsatellite markers. Hereditas, 28(12), 1549-1554.). We state it not clear and inaccurate in our former manuscript. We delete the sentence to avoid confusing the reader. (Line 237-239)

Point 23: Line 203. Provide a statement on the full names of all these statistics at the end of the table.

Response 23: We have added the full names of all these statistics at the end of the table. (Line 241-243)

Point 24: Line 213. State the significant level is enough. You may write as: both of the sign and Wilcoxon tests showed significant deviations from the mutation-drift balance of the population. By the way, what does this deviation mean? A sign of bottleneck? Please explain it.

Response 24: We have changed the sentence to ‘both of the sign and Wilcoxon tests showed significant deviations from the mutation-drift balance of the population’ (Line 252-254) and made some revision in other place (Line 341-342). We used the sign test and Wilcoxon test to test three mutation models, if these models showed significant deviations in the test, then it can be considered that there is significant excess heterozygosity in the population, that is, the rate of heterozygosity decrease is lower than the rate of allele loss in the population, suggesting the bottleneck. However, TPM model is more suitable for microsatellite data according to former studies. In our test, TPM model showed no significant deviations in the two test, indicating that the L. leishanense population has not recently experienced a bottleneck (Line 339-353).

Point 25: Line 244. Change ‘who’ to ‘which’.

Response 25: We have revised it in the manuscript. (Line 285)

Point 26: Lines 260-261. This sentence makes no sense, I suggest deleting it.

Response 26: We have deleted it in the manuscript. (Line 308-309)

Point 27: Lines 278-280. I was confused, so you used two methods to evaluate inbreeding? What is the exact inbreeding level of this species?

Response 27: We used Fis value to evaluate inbreeding. Now we changed our statement to ‘Although Fis value of the population is on the low degree, this does not mean that there is no inbreeding between the individuals in the population. According to our year-by-year field work, its population size is declining. This undoubtedly increases the possibility of its inbreeding. Inbreeding has a negative effect on the fitness of the pop-ulation, including fertility and viability, which is not conducive to the long-term development of the population.’ and added some reasonable explanations like ‘We could not find more obvious molecular evidence of inbreeding may be due to our restricted sampling size and the relatively high number of alleles found. As with high number of alleles the probability of getting homozygote hgenotypes in one locus is very low. Thus, it will influence our detection of inbreeding.’ in ‘4.2’ (Line 324-332).

Point 28: Line 310. Delete ‘extremely’.

Response 28: We have deleted it in the manuscript. (Line 367)

Point 29: In addition, I suggest that the author can try to use MsVar to analyze population dynamics, and there may be more substantial results.

Response 29: We do try to use MsVar to analyze population dynamics, unfortunately, all the link we find are broken (Beaumont, M. A. (1999). Detecting population expansion and decline using microsatellites. Genetics, 153(4), 2013-2029.; Storz, J. F., & Beaumont, M. A. (2002). Testing for genetic evidence of population expansion and contraction: an empirical analysis of microsatellite DNA variation using a hierarchical Bayesian model. Evolution, 56(1), 154-166.). We try to contact with the author, but still haven't heard back.

Instead, we used three mutation models and allele frequency distribution to test whether the population has experienced bottleneck and then we estimated its effective population size. Further, we increased calculating the mean relatedness of the sampling individuals for every year and pairwise year Fst values. These results can only describe its partial and incomplete population dynamics. Although as not obtaining the substantial results as MsVar, the results we get can still guide our field work for its conservation. Nevertheless, expanding our sample size and using more programs to obtain accurate and substantial results are the direction of our working. Below, we simply state the method we used to test population bottleneck.

We test three mutation models to measure whether the population exist significant excess heterozygosity. If so, the rate of heterozygosity decrease is lower than the rate of allele loss in the population, suggesting the bottleneck. Among these models, infinite allele model (IAM) showed significant deviations, both stepwise mutation model (SMM) and two-phased model of mutation (TPM) showed no significance. But IAM assumes that there is only one mutation of an allele in a population, and each mutation produces a new allele, which is generally used in isozyme or DNA sequencing data. TPM model is more suitable for microsatellite data according to former studies. So we chose to accept the results of TPM that the population did not experience bottleneck recently. Further, we check the allele frequency distribution results, if the population has experienced bottleneck recently, the distribution of alleles with low frequency will change to a mid-frequency distribution, thus, the allele frequency distribution will deviate from ‘L’ type. But our results showed a typical ‘L’ type, indicating the population did not experience bottleneck, too. (Line 339-364)

Best regards,

Prof. Wu Hua

Reviewer 3 Report

RE: Genetic Diversity and Population Dynamics of Leptobrachium 2 leishanense (Anura: Megophryidae) as Determined by Tetranu-3 cleotide Microsatellite Markers Developed from its Genome

by Chao Fu et al.

This paper describe a population genetic study using microsatellites (which is a technique which is becoming obsolete as SNPs studies are now used in the major part of populationgenetics studies......therefore the paper is an average study, but the fact that it is dealing with amphibians which are  severily endangered due to climate changes, I think this paper have some merits.

However the authors do not find any evidence of inbreeding and I think that this is due to the low sample size and the relatively high number of alleles found....as with high number of alleles the probability of getting homozygote hgenotypes in one locus is very low....

the authors decided to pool all samples 2012, 2013, 2014, 2015 and 2018, collecting 24 individuals per year....if the population is fluctuating in size it could have been interesting to not pool the samples and analyse the population s for every year and estimate the FST between years....I guess there is the possibility to find significant FST values given the fact that evidences of bottleneck have been found.

I am aware that the authors have estimated HWE for every years but it would be nice to write the FIS values when they have significant deviations....

The authors mentioned that probably  some individuals in the same family have been collected....therefore it would have been interesting if the mean relatedness of the individuals for every year can be calculated...it can be domne in GENALEX

Author Response

Dear reviewer,

Thanks for your kindly comments and suggestions. We have tried our best to rewrite and improve our manuscript and streamlined the manuscript in this version. Below, we list your comments verbatim along with our response in bold, detailing how we addressed the suggestions and criticisms.

Point 1: The authors do not find any evidence of inbreeding and I think that this is due to the low sample size and the relatively high number of alleles found....as with high number of alleles the probability of getting homozygote hgenotypes in one locus is very low.

Response 1: Thanks for your helpful suggestions. We have added your suggestions and revised our statement in ‘4.2’ like ‘Although Fis value of the population is on the low degree, this does not mean that there is no inbreeding between the individuals in the population. According to our year-by-year field work, its population size is declining. This undoubtedly increases the possibility of its inbreeding. Inbreeding has a negative effect on the fitness of the population, including fertility and viability, which is not conducive to the long-term development of the population. We could not find more obvious molecular evidence of inbreeding may be due to our restricted sampling size and the relatively high number of alleles found. As with high number of alleles the probability of getting homozygote hgenotypes in one locus is very low. Thus, it will influence our detection of inbreeding.’. (Line 324-337)

Point 2: The authors decided to pool all samples 2012, 2013, 2014, 2015 and 2018, collecting 24 individuals per year....if the population is fluctuating in size it could have been interesting to not pool the samples and analyse the populations for every year and estimate the FST between years....I guess there is the possibility to find significant FST values given the fact that evidences of bottleneck have been found.

Response 2: We do use three mutation models and allele frequency distribution to test whether the population has experienced bottleneck. But the results show the population has not experienced bottleneck. We have tried to compare per year Fst values according to your suggestion, but we have not found significant difference between years. ‘(Line 139-140, Line 217, Line 222-224)’. Below, we simply state the method we used to test population bottleneck.

We test three mutation models to measure whether the population exist significant excess heterozygosity. If so, the rate of heterozygosity decrease is lower than the rate of allele loss in the population, suggesting the bottleneck. Among these models, infinite allele model (IAM) showed significant deviations, both stepwise mutation model (SMM) and two-phased model of mutation (TPM) showed no significance. But IAM assumes that there is only one mutation of an allele in a population, and each mutation produces a new allele, which is generally used in isozyme or DNA sequencing data. TPM model is more suitable for microsatellite data according to former studies. So we chose to accept the results of TPM that the population did not experience bottleneck recently. Further, we check the allele frequency distribution results, if the population has experienced bottleneck recently, the distribution of alleles with low frequency will change to a mid-frequency distribution, thus, the allele frequency distribution will deviate from ‘L’ type. But our results showed a typical ‘L’ type, indicating the population did not experience bottleneck, too. (Line 339-377)

Point 3: I am aware that the authors have estimated HWE for every years but it would be nice to write the FIS values when they have significant deviations.

Response 3: We have revised it in the manuscript. (Line 202-212)

Point 4: The authors mentioned that probably some individuals in the same family have been collected....therefore it would have been interesting if the mean relatedness of the individuals for every year can be calculated...it can be done in GENALEX.

Response 4: We have calculated it in our revised manuscript. (Line 139-140, Line 218-221, Line 224-227, Line 300-301)

Best regards,

Prof. Wu Hua
